# Gaussian-Based Adaptive Fish Migration Optimization Applied to Optimization Localization Error of Mobile Sensor Networks

**DOI:** 10.3390/e24081109

**Published:** 2022-08-12

**Authors:** Yong Liu, Wei-Min Zheng, Shangkun Liu, Qing-Wei Chai

**Affiliations:** 1College of Ocean Science and Engineering, Shandong University of Science and Technology, Qingdao 266590, China; 2Laboratory for Marine Geology, Qingdao National Laboratory for Marine Science and Technology, Qingdao 266237, China; 3College of Computer Science and Engineering, Shandong University of Science and Technology, Qingdao 266590, China

**Keywords:** heuristic algorithms, fish migration optimization, localization, mobile sensor networks, monte carlo localization

## Abstract

Location information is the primary feature of wireless sensor networks, and it is more critical for Mobile Wireless Sensor Networks (MWSN) to monitor specific targets. How to improve the localization accuracy is a challenging problem for researchers. In this paper, the Gaussian probability distribution model is applied to randomize the individual during the migration of the Adaptive Fish Migration Optimization (AFMO) algorithm. The performance of the novel algorithm is verified by the CEC 2013 test suit, and the result is compared with other famous heuristic algorithms. Compared to other well-known heuristics, the new algorithm achieves the best results in almost 21 of all 28 test functions. In addition, the novel algorithm significantly reduces the localization error of MWSN, the simulation results show that the accuracy of the new algorithm is more than 5% higher than that of other heuristic algorithms in terms of mobile sensor node positioning, and more than 100% higher than that without the heuristic algorithm.

## 1. Introduction

With the efforts of more and more researchers, many technologies have become mature and inexpensive, such as communication theory, micro electromagnetic systems, and integrated circuits. Based on these technologies, wireless sensor networks (WSNs) have been widely used, and their performance has been dramatically improved [1]. A sensor node can collect valuable information, process it, and pass it to another sensor node, and finally, this information reaches the sink node [2]. The location information of sensor nodes is essential for WSNs to ensure that users match the collected data and monitoring targets and make correct decisions [3]. Location information is usually provided by the Global Positioning System (GPS), but only a few sensor nodes are equipped with GPS due to cost and energy constraints. Sensor nodes with GPS are called anchor nodes, and other nodes are called unknown nodes because their location is unknown [4].

In a static wireless sensor network, the sensor nodes are set in a certain position and will not change, the position of the unknown node can be estimated by many algorithms based on the information of the anchor node. These algorithms are divided into two categories according to whether they rely on the distance information between sensor nodes, namely range-based localization algorithms and range-free localization algorithms [5,6]. Some range-based localization algorithms represented in this section. For an unknown node, the time of arrival (TOA) from different stations can be used to estimate its location [7]. In [8], for an unknown node, the time of arrival (TOA) from different stations can be used to estimate its location. The authors exploit the time difference of arrival (TDOA) information to estimate the location of unknown nodes and employ continuous unconstrained minimization and a generalized trust-region subproblem to optimize the problem. In the Received Signal Strength Indication (RSSI) algorithm, the distance between them can be calculated based on the signal attenuation between sensor nodes [9]. In range-based localization algorithms, linear distance or direction information between sensor nodes is usually utilized. This class of algorithms can provide more accurate location information but requires additional components to obtain distance or direction. Therefore, the economic cost and energy cost are not ideal [10]. The range-free localization algorithm can solve the wireless sensor network localization problem only with a more straightforward sensor node than the range-based localization algorithm. The weighted centroid localization (WCL) algorithm only utilizes the signal strength to estimate the location of the unknown node; this machine ensures that WSNs can work in a complex deployment environment [11]. The DV-Hop localization algorithm calculates the distance from the anchor node to the unknown node according to the distance of each hop of the anchor node [12]. Chen et al. introduce the different calculations about the distance between anchor nodes and unknown nodes. They utilize the average hop-size of all anchor nodes to estimate the location of unknown nodes rather than each anchor node with its hop-size [13]. In [14], the authors propose an Ad hoc Positioning System (APS) method to reduce positioning errors, which combines propagation and GPS triangulation information to estimate the location of unknown nodes.

There is a serious challenge in the positioning of mobile sensor nodes in MWSN; that is, the positioning error is huge, and with the movement of mobile sensor nodes, the positioning error will become larger and larger. To address this problems, this paper introduces a novel heuristic algorithm. Heuristic algorithm is a powerful tool to solve many engineering problems. Some scholars use the excellent performance of the heuristic algorithm in optimization to reduce positioning error. The adaptive strategy is combined with a compact Particle Swarm Optimization (PSO) algorithm, and this algorithm can run on a memory limitation device. Simulation results indicate that localization error is significantly reduced [15]. PSO algorithm is used to enhance the localization accuracy-based distance between sensor nodes that RSSI obtained [16]. The performance of general heuristics applied to WSN localization is compared in [17]. Some researchers work on the localization of sink nodes. In [18], the authors proposed a method that utilizes the Grey Wolf Optimization (GWO) algorithm to find the location of the sink node. In [19], a Compact Black Hole (CBH) algorithm is introduced and applied to solve the localization of mobile sensor node problem.

With the increasing attention of scholars, there are many excellent novel or improved heuristic algorithms. In previous decades, only some basic and simple heuristic algorithms were proposed and used, such as Genetic Algorithm (GA) [20], PSO algorithm [21], Ant Colony Optimization (ACO) algorithm [22], Whale Optimization Algorithm (WOA) [23], and Artificial Bee Colony (ABC) algorithm [24]. In recent years, scholars have proposed various heuristic algorithms inspired by natural phenomena or swarm intelligence action. The Black Hole (BH) algorithm mimics a black hole in nature, where the matter around it is devoured [25]. If the individual is too close to the global best candidate solution in the BH algorithm, it will be randomly initialized. Chu et al. proposed a PSO-based Cat Swarm Optimization (CSO) algorithm, in which the authors introduced two models: a finding model and a tracking model. According to the cooperation of these two models, the algorithm performs well in complex optimization problems [26]. In [27], four novel transformation functions are applied to the Binary Grey Wolf Optimization (BGWO) algorithm, which outperforms traditional BGWO on feature selection problems. The multi-surrogate strategy efficiently improves the convergence rate of binary PSO when facing complex multi-dimensional problems [28]. Useful information from the optimization process can be reused, which can further guide the movement of the population. In [29], six information feedback models are introduced, and the experimental results show that this strategy can improve the search performance of the heuristic algorithm. Gao et al. proposed a novel Difference Evolutionary (DE) algorithm to solve the job-shop scheduling problem [30], which adopted a novel selection mechanism and significantly enhanced the global search ability of DE. Adaptive parameters are used to limit the movement of the Substance Search (SMS) algorithm, and the new algorithm is applied to hide watermarks into QR codes [31].

## 2. Related Work

In order to reduce the localization error of mobile sensor nodes, this paper combines the Sequential Monte Carlo Localization (SMCL) method and heuristic algorithm. The reason for using the heuristic algorithm is that the optimal value can be quickly calculated, which can ensure the timely positioning of the position of the mobile sensor node. This section briefly presents the mechanism of the SMCL method and AFMO algorithm.

### 2.1. Adaptive Fish Migration Opmtimization Algorithm

AFMO algorithm was proposed in 2020, and is a modified version of the Fish Migration Optimization (FMO) algorithm. The FMO algorithm mimics the whole life course of fish and divides the life of fish into five stages. There are many accidents during fish growth, so many individuals cannot grow up safely. In addition, these fish would return to their birthplace when adults, producing offspring. Therefore, the survival rate is introduced by authors in FMO [32], and they are set at 5%, 10%, and 100% in stage 3, stage 4, and stage 5, respectively. In the FMO algorithm, the energy of the individual increases with the number of iterations, and when the individual’s energy exceeds a particular value, the individual will enter the next stage. When individuals return to their birth positions or die, new individuals are randomly generated to keep the population size unchanged. This scheme ensures that the FMO algorithm performs strongly in avoiding local optima. However, the algorithm has poor searchability in the single-modal test function because the exploitation ability is weak.

In the AFMO algorithm, as Figure 1 shows, the life of fish consists of four stages, and the survival rate is 15%, 35%, and 100% in stage 2, stage 3, and stage 4. Some studies have shown that a suitable parameter adjustment strategy can balance exploration and exploitation to enhance the optimization performance in the heuristic algorithms [33]. The AFMO algorithm introduced a novel strategy to adjust the energy update of the FMO algorithm, and the detail is presented in the following:(1)$$Ene_{i}^{t+1}=Ene_{i}^{t}+r_{e}\cdot Ene_{max}\cdot \frac{fit_{i}-fit_{best}}{fit_{max}-fit_{best}}$$
where the $Ene_{i}^{t}$ is the energy of the *i*-th individual at t iteration, $Eng_{max}$ is a constant value and set at 200 in [34]. To enhance the diversity of the population, a perturbation element $r_{e}$ is added to Equation (Equation 1), which is a random value between 0.2 and 0.6. The fitness function would evaluate the individual of AFMO, and the fitness value of the *i*-th individual is represented by $fit_{i}$. The fitness values of the best and worst individuals are represented by $fit_{best}$ and $fit_{max}$. This mechanism promotes individuals with poor fitness values to the next stage and makes it initialized with greater probability. The energy not only determines if the individual grows up to the next stage but also influences the individual’s update at one iteration. The detail of the update is shown in Equation (Equation 2).
(2)$$X_{i}^{t+1}=X_{i}^{t}+w\cdot \left(X_{best}^{t}-X_{i}^{t}\right)\cdot \frac{Ene_{i}^{t}}{Ene_{max}}+\frac{fit_{i}-fit_{r}}{|fit_{i}-fit_{r}|}\cdot RC\cdot \left(X_{i}^{t}-X_{r}^{t}\right)$$
where $X_{i}^{t}$ is the position of the *i*-th individual at the *t* iteration, and *w* is a parameter that controls the individual’s range of motion, which is a variable value that decreases from 2 to 0.4 during the operation of the AFMO algorithm. $X_{best}^{t}$ is the position of the individual with the best fitness value, $X_{r}^{t}$ and $fit_{r}$ are the position and fitness value of a randomly selected individual from the population. The AFMO algorithm adds a learning strategy to the FMO algorithm, randomly selects an individual as the learning object, and compares it with the *i*-th individual. If the *i*-th individual is worse than the learning object, it is close to the learning object, and vice versa. The RC is a random number between 0 and π/10, and it can adjust the study strength of the algorithm.

Although the AFMO algorithm enhances the performance of the original FMO algorithm and obtained better results than other famous heuristic algorithms in the CEC 2013 test suit, it has disadvantages in unimodal optimization problems. This paper introduces the novel algorithm called the Gaussian-Based Adaptive Fish Migration Optimization (GAFMO) algorithm, which applies the Gaussian distribution model to the migration process of AFMO. This mechanism enhances the population diversity in the migration process of fish and ensures the exploitation ability.

### 2.2. Sequential Monte Carlo Localization Method

The authors introduced the SMCL method to enhance the localization accuracy of mobile sensor nodes of WSNs [35]. Twenty years ago, there was little research on the localization of mobile sensor nodes, but similar problems were widely studied in robotics. Researchers usually estimate the robot’s position in robot localization based on measurement models and observational data. The measurement model is built from previously collected data, and the model is continuously updated during robot operation. If the measurement model and observational data obey a Gaussian distribution, robot localization can be solved by using a Kalman filter [36]. In some cases, the Kalman filter can not be used when the problem is non-Gaussian; the Markov localization method is introduced [37].

Sensor node localization has different challenges to solve than robot localization: 1. Sensor nodes are placed on an unknown map or terrain. 2. The speed or direction of the mobile sensor node cannot be obtained. 3. Mobile sensor nodes do not have enough energy and memory to estimate localization by integrating information collected by other sensor nodes [38]. In [35], based on the current location information, the authors try to obtain the probability distribution of the possible locations of the mobile sensor nodes at the next time point. However, there are so many possible locations that it is difficult to estimate the actual location, and existing location information becomes inaccurate over time. If the speed is a random value between 0 and $V_{max}$, and the direction of mobile sensor nodes is unknown, the probability distribution can be presented in the following:(3)$$P_{i}^{t}=\left\{\begin{array}{cc} \frac{1}{\pi V_{max}^{2}} & \;if\;abs\left(P_{i}^{t}-P_{i}^{t-1}\right)\geq V_{max}\\ 0 & \;otherwise \end{array}\right.$$
The SMCL method introduces a filtering mechanism based on new observations from other sensor nodes to exclude impossible locations. There are four situations for sensor node localization in MWSN: outsiders, arrivals, leavers, and insiders. When a sensor node is not heard at the current and previous time point, it belongs to outsiders; if a sensor node is not heard at the previous time point but is heard at the current time, it is in arrivals; if the sensor node is not heard at the current time point, is heard at the previous time point, it belongs to the leavers; if the sensor node is heard at the previous time point and the current time point, it is an insiders. These situations are presented in Figure 2, and A, B, C, and D represent levers, insiders, arrivals, and outsiders, respectively. The circle filled with blue is the sensor range of the node at the *t* − 1 time point, and the circle filled with yellow is the sensor range of the node at the *t* time point.

Arrivals and leavers provide the most helpful information for the localization of a mobile sensor node as it is located around the communication boundary of the arrivals or leavers. For the cases of an outsider, the information of mobile sensor nodes can be transmitted to the outsider node by the neighbor nodes. The detail of this process is shown in Figure 3. Although the outsider can not hear the information of the mobile sensor node, it can be regarded as leavers or arrivals of mobile sensor nodes within a 2R radius. Insiders cannot locate outside the radius of the mobile sensor node.

## 3. Gaussian-Based Adaptive Fish Migration Optimization

In nature, the growth of fish is accompanied by a variety of adverse factors such as disease, food scarcity, and predators that prevent so many people from reaching adulthood. To simulate this phenomenon, the authors introduced a survival rate mechanism that maintains population size by randomly generating new individuals [32]. Although this method ensures the diversity of the population and the ability to avoid falling into the local optimum, it leads to the weak performance of the algorithm on the unimodal problem. In this paper, the Gaussian probability distribution model is introduced to generate new individual migration processes of AFMO and is presented in Figure 4.

Figure 4a shows the results of 3000 iterations of a Gaussian function with parameters μ of 0 and σ of 16. Each point in the graph is generated by the Gaussian function in one iteration and they are linked. We can see that the output of the Gaussian function is between −20 and 20 in most cases, and the maximum absolute value is about 50. The distribution of Gaussian function is shown in Figure 4b, the output is located in the range between −16 and 16, with a 68.27% probability, and in the range between −32 and 32 with 95.45% probability. In the heuristic algorithm, if a new individual is generated by this Gaussian probability distribution model, it will be within 32 units of u in most cases. This model is applied to migration processes of AFMO to enhance the exploitation ability and the detail is shown as the following:(4)$$X_{mig}^{t+1}=Gaussian\left(X_{best}^{t},\sigma \right)+\frac{fit_{i}-fit_{r}}{fit_{i}-fit_{best}}\cdot \left(X_{i}^{t}-X_{r}^{t}\right)$$
where the $X_{mig}^{t+1}$ represents the individual after migration at t+1 iterations, $X_{best}^{t}$ is the individual with optimal fitness value at *t* iterations, and the σ is set at 16 in this article. The $fit_{i}$, $fit_{r}$, and $fit_{best}$ are the fitness values of the *i*-th individual, randomly selected individual, and best individual. The positions of the *i*-th individual and randomly selected individual at *t* iterations are represented by $X_{i}^{t}$ and $X_{r}^{t}$. This equation ensures that new individuals are generated in promising regions (near the best individuals), so it can find better candidate solutions with greater probability. Furthermore, new individuals are attracted to another randomly selected individual, and the better the randomly selected individual, the stronger the attraction. The detail of the new algorithm is shown in Algorithm 1.
**Algorithm 1:** The Gaussian-Based Adaptive Fish Migration Algorithm.
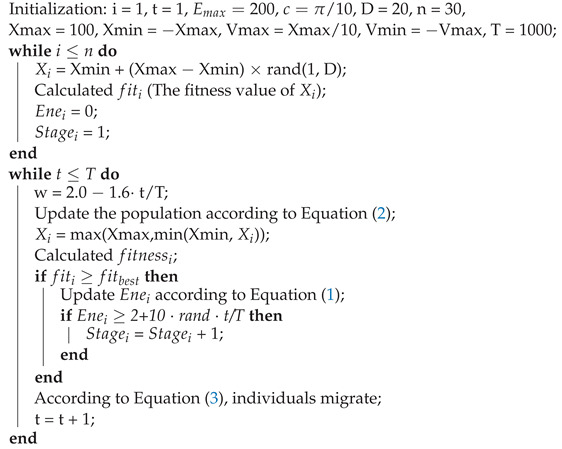


## 4. Experimental Results and Discussion

In this section, a comparison of the new algorithm under the CEC 2013 test suite and locating mobile sensor nodes with other well-known algorithms is presented. The results provided by the CEC 2013 test suite illustrate the comprehensive performance of the new algorithm, which shows that the new algorithm has excellent optimization capabilities over other well-known heuristic algorithms. Localization simulation experiments of mobile sensor nodes can demonstrate the performance of the new algorithm in solving specific problems in the real world. The experiments were completed with Matlab 2020a on a personal computer with Intel Core i7-10700k (5.1 GHz) and 48 G memory, and all experiments were processed under the same parameters, such as population size, dimensions, or iterations.

### 4.1. Experiments under CEC 2013 Test Suite

The CEC 2013 test suite is proposed to estimate the performance of heuristic algorithms on single-objective optimization problems, which are the basis of niche, multi-objective, and constrained optimization algorithms. This paper tests heuristic algorithms on 28 test functions of the CEC 2013 test suite to fully and fairly verify the new algorithm’s performance. The test functions were separated into three classes, which are Unimodal Functions ($f_{1}$ to $f_{5}$), Basic Multimodal Functions ($f_{6}$ to $f_{20}$), and Composition Functions ($f_{21}$ to $f_{28}$); all functions are minimization problems. The novel algorithm is compared with the classical heuristic algorithm PSO, the original FMO, and the WOA and BH algorithms proposed in recent years. The parameter setting is shown in [23,25,34,39]. The experimental results are shown in the tables below, and the algorithms were used to find the optimal solution for each test function in 20, 30, and 40 dimensions. All results are the mean and standard deviation of 48 runs.

Various test functions can verify the different performances of heuristic algorithms. In order to compare the exploitation ability of algorithms, uni-modal test functions are introduced in CEC 2013. It has only one optimal solution in a limited area, so the heuristic algorithms with strong exploitation ability can obtain great candidate solutions. The experimental results under uni-modal test functions are presented in Table 1, and the novel algorithm gets the best results in all uni-modal test functions for each dimension except $f_{4}$. In all uni-modal test functions, the novel algorithm performance was excellent in six standard deviation results, which shows that the novel algorithm has excellent stability.

The combination function combines uni-modal and multi-modal test functions, which can verify the comprehensive performance of the heuristic algorithm. The GAFMO algorithm obtains the best results in almost all experiments, as shown in Table 2, instead of $f_{21}\left(40\right)$, $f_{23}\left(30\right)$, $f_{26}\left(20\right)$, and $f_{26}\left(30\right)$. This phenomenon indicates that the novel algorithm has a strong exploitation ability and an ability to avoid optimal local values. Like the other experiments, the novel algorithm has excellent composition functions and stability.

Since the new algorithm introduces a Gaussian probability distribution model based on AFMO, it has a stronger exploration performance than AFMO, which can be proved by the experimental results presented in Table 3. In multi-modal test functions, the novel algorithm obtains the greatest result at $f_{6}$, $f_{7}$, $f_{9}$ to $f_{14}$, and $f_{19}$ for each dimension. As the dimension increases, the new algorithm performs better and better on the $f_{15}$ and $f_{18}$ test functions and achieves the best score among the five algorithms in the case of 40 dimensions. The experimental data shows that the new algorithm has an excellent performance in solving high-dimensional and high-complexity problems. The $f_{8}$ test function is not discussed in this article because the algorithms under this function perform similarly and provide no useful information. In addition, the new algorithm has the lowest standard deviation of the 25 results (equivalent to 55% of all multi-modal experimental results), which means that it can obtain a solution closer to the mean shown in Table 3 than other algorithms in most cases.

### 4.2. Experiments under CEC 2013 Test Suite

The CEC 2013 test suite is proposed to estimate the performance of heuristic algorithms on single-objective optimization problems, which are the basis of niche, multi-objective, and constrained optimization algorithms. This paper tests heuristic algorithms on 28 test functions of the CEC 2013 test suite to fully and fairly verify the new algorithm’s performance. The test functions are separated into three classes which are Unimodal Functions ($f_{1}$ to $f_{5}$), Basic Multimodal Functions ($f_{6}$ to $f_{20}$), and Composition Functions ($f_{21}$ to $f_{28}$), and all of these functions are minimization problems. The novel algorithm is compared with the classical heuristic algorithm PSO, the original FMO, and the WOA and BH algorithms proposed in recent years. The parameter setting is shown in [23,25,34,39]. The experimental results are shown in the tables below, and the algorithms were used to find the optimal solution for each test function in 20, 30, and 40 dimensions. All results are the mean and standard deviation of 48 runs.

Various test functions can verify the different performances of heuristic algorithms. In order to compare the exploitation ability of algorithms, uni-modal test functions are introduced in CEC 2013. It has only one optimal solution in a limited area, so the heuristic algorithms with strong exploitation ability can obtain great candidate solutions. The experimental results under uni-modal test functions are presented in Table 1, and the novel algorithm gets the best results in all uni-modal test functions for each dimension except $f_{4}$. In all uni-modal test functions, the novel algorithm performance was excellent in six standard deviation results, which shows the novel algorithm has excellent stability.

The composition function consisted of uni-modal and multi-modal test functions, which can verify the comprehensive performance of the heuristic algorithm. The GAFMO algorithm obtains the best results of almost all experiments, as shown in Table 2, instead of $f_{21}\left(40\right)$, $f_{23}\left(30\right)$, $f_{26}\left(20\right)$, and $f_{26}\left(30\right)$. This phenomenon indicates that the novel algorithm has a strong exploitation ability and ability to avoid optimal local value. Like the other experiments, the novel algorithm has excellent composition functions and stability.

Since the new algorithm introduces a Gaussian probability distribution model based on AFMO, it has a stronger exploration performance than AFMO, which can be proved by the experimental results presented in Table 3. In multi-modal test functions, the novel algorithm obtains the greatest result at $f_{6}$, $f_{7}$, $f_{9}$ to $f_{14}$, and $f_{19}$ for each dimension. As the dimension increases, the new algorithm performs better and better on the $f_{15}$ and $f_{18}$ test functions and achieves the best score among the five algorithms in the case of 40 dimensions. The experimental data shows that the new algorithm has an excellent performance in solving high-dimensional and high-complexity problems. The $f_{8}$ test function is not discussed in this article because the algorithms under this function perform similarly and provide no useful information. In addition, the new algorithm has the lowest standard deviation of the 25 results (equivalent to 55% of all multi-modal experimental results), which means that it can obtain a solution closer to the mean shown in Table 3 than other algorithms in most cases.

### 4.3. Simulation Experiments of Localization of MWSN

In this section, heuristic algorithms are used to reduce the localization error of the SMCL method. The individual of heuristic algorithms represent a candidate position of the mobile sensor node. The optimal position is found by iteration of the algorithm; that is, the most probable position in the promising area. Through these simulation experiments, the performance of the heuristic algorithm to solve real problems can be verified. Experiments are performed under different conditions, such as the number of anchor nodes, sensor nodes, and the communication radius, but the deployment area is 200 m × 200 m for all experiments. The maximum speed of a mobile sensor node is its communication radius. The new algorithm is compared with the PSO, BH, and WOA algorithms, and the detailed results of these experiments are shown in the table below, with the best results for each experiment are marked in bold.

In Table 4, the experiment is performed with different anchor node number, the number of sensor node is set at 200, and the communication radius is 30 m. The results revel that the heuristic algorithm can significantly enhance the localization accuracy of SMCL; specifically, the novel algorithm can obtain better results than other heuristic algorithms.

The number of sensor nodes is the variable in Table 5, and the constant elements are the number of anchor nodes and the communication radius, which are 10 and 30 m, respectively. The more sensor nodes, the more complex the sensor node topology, but the mobile sensor node can receive more anchor node information because it is connected to more sensor nodes. In this simulation experiment, the new algorithm reduces the positioning error by more than 30% compared with the original SMCL method. Compared to other algorithms, the new algorithm works best. The communication radius determines how many other nodes a sensor node can communicate with. As the communication radius increases, the messages broadcast by the mobile sensor nodes can be received by more anchor nodes, so the localization is more accurate. The results shown in Table 6 are obtained with different communication radii, 200 sensor nodes, and 15 anchor nodes.The results show that the new algorithm has excellent optimization performance in the positioning of mobile sensor nodes in MWSN, and the optimization ability is significantly improved compared with other heuristic algorithms.

## 5. Conclusions

This paper analyzes the feature and performance of AFMO, which has an excellent performance in multimodal problems, but the strong exploration ability limits the local search ability. This means the AFMO can not obtain satisfactory results in unimodal problems. In order to enhance the exploitation performance, we introduce the Gaussian probability distribution to the migration process of AFMO. This mechanism ensures that the novel algorithm obtains better results in unimodal problems and retains the original exploration ability. The performance of the new algorithm is verified by the CEC 2013 test suit, and the experimental results show that the novel algorithm has better exploitation performance and a solid ability to avoid the optimal local value. The new algorithm achieves the 60 best results in all 84 experiments; that is, the new algorithm wins in 71.4% of the experiments. In addition, this paper applies the heuristic algorithm to solve the localization of mobile sensor nodes in MWSN. The simulation experiments reveal that the heuristic algorithm can significantly enhance the localization accuracy of mobile sensor nodes. Specifically, the new algorithm can improve the localization accuracy of mobile sensor nodes by more than 5% compared to other heuristic algorithms. This technique can also solve the localization of the robot in the room, the robot can provide more information to the localization system but there is more problems to solve than MWSN. This paper proves that the Gaussian probability distribution model can enhance the exploitation ability and not reduce the exploration ability. This model can apply other algorithms to further improve the ability of heuristic algorithms. In addition, other probability modes have their own features, and they may be more suitable for enhancing the performance of heuristic algorithms or solving localization problems. This is interesting work to do.

## Figures and Tables

**Figure 1 entropy-24-01109-f001:**
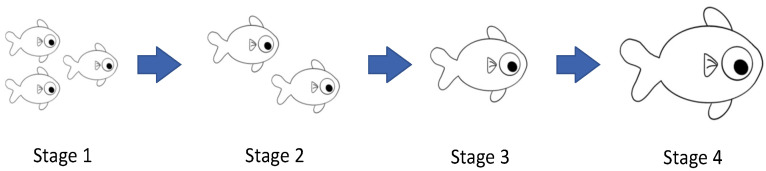
The grow-up process of fish.

**Figure 2 entropy-24-01109-f002:**
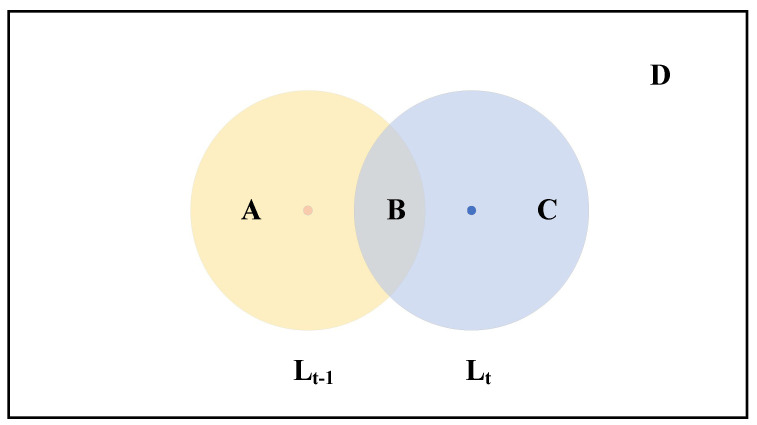
Sensor nodes movement.

**Figure 3 entropy-24-01109-f003:**
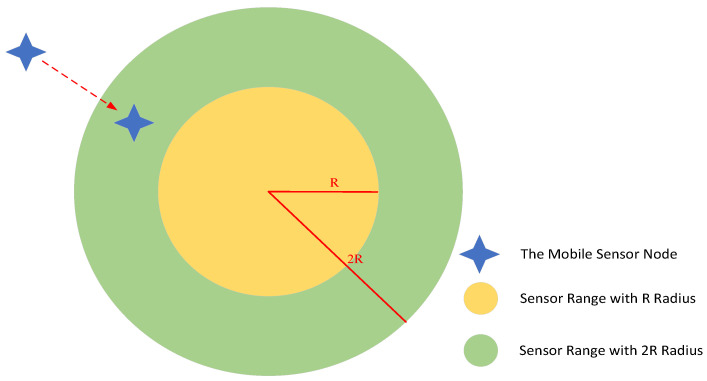
Sensor nodes movement in case of outsider.

**Figure 4 entropy-24-01109-f004:**
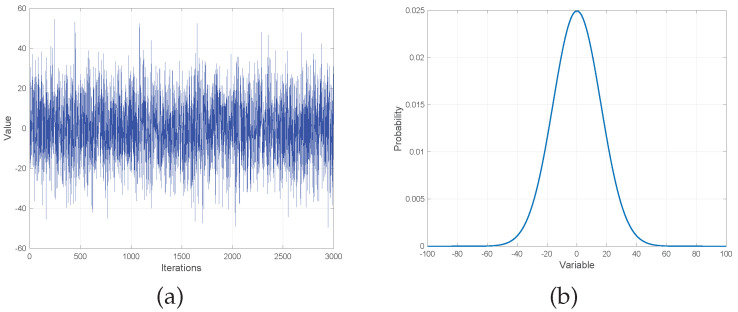
Results of running the Gaussian function (**a**) and the distribution of Gaussian function (**b**) with μ = 0 and σ = 16.

**Table 1 entropy-24-01109-t001:** The experimental results under uni-modal test functions.

Algorithm		PSO		AFMO		WOA		BH		GAFMO	
	Dim	Ave	Std	Ave	Std	Ave	Std	Ave	Std	Ave	Std
$f_{1}$	20	$-1.30\times 10^{3}$	$3.61\times 10^{2}$	$-1.26\times 10^{3}$	$2.01\times 10^{2}$	$-1.27\times 10^{3}$	$9.84\times 10^{1}$	$5.62\times 10^{3}$	$9.60\times 10^{2}$	** $-1.40\times 10^{3}$ **	** $2.49\times 10^{1}$ **
	30	$-1.15\times 10^{3}$	$4.24\times 10^{2}$	$-6.72\times 10^{2}$	$7.22\times 10^{2}$	$-6.57\times 10^{2}$	$3.73\times 10^{2}$	$1.65\times 10^{4}$	$1.39\times 10^{3}$	$-1.40\times 10^{3}$	$6.25\times 10^{1}$
	40	$-8.50\times 10^{2}$	$8.01\times 10^{2}$	$2.10\times 10^{3}$	$3.21\times 10^{3}$	$1.03\times 10^{3}$	$9.40\times 10^{2}$	$2.56\times 10^{4}$	$2.16\times 10^{3}$	$-1.40\times 10^{3}$	$1.70\times 10^{2}$
$f_{2}$	20	$2.30\times 10^{6}$	$1.94\times 10^{6}$	$3.13\times 10^{6}$	$3.01\times 10^{6}$	$4.60\times 10^{7}$	$2.24\times 10^{7}$	$2.90\times 10^{7}$	$3.63\times 10^{6}$	$1.93\times 10^{6}$	$3.11\times 10^{6}$
	30	$1.03\times 10^{7}$	$6.62\times 10^{6}$	$7.09\times 10^{6}$	$2.76\times 10^{6}$	$1.17\times 10^{8}$	$5.14\times 10^{7}$	$1.62\times 10^{8}$	$2.37\times 10^{7}$	$5.61\times 10^{6}$	$6.51\times 10^{6}$
	40	$1.67\times 10^{7}$	$5.53\times 10^{6}$	$1.92\times 10^{7}$	$8.82\times 10^{6}$	$1.48\times 10^{8}$	$3.95\times 10^{7}$	$9.29\times 10^{7}$	$8.58\times 10^{6}$	$1.01\times 10^{7}$	$1.09\times 10^{7}$
$f_{3}$	20	$3.02\times 10^{9}$	$4.23\times 10^{9}$	$1.05\times 10^{9}$	$2.58\times 10^{9}$	$1.01\times 10^{11}$	$2.75\times 10^{11}$	$2.38\times 10^{15}$	$5.00\times 10^{15}$	$1.00\times 10^{8}$	$1.51\times 10^{8}$
	30	$1.16\times 10^{10}$	$1.31\times 10^{10}$	$1.04\times 10^{10}$	$1.06\times 10^{10}$	$5.95\times 10^{10}$	$4.31\times 10^{10}$	$6.82\times 10^{15}$	$1.27\times 10^{16}$	$1.02\times 10^{9}$	$2.45\times 10^{9}$
	40	$1.53\times 10^{10}$	$1.31\times 10^{10}$	$1.03\times 10^{10}$	$5.52\times 10^{9}$	$7.88\times 10^{10}$	$3.66\times 10^{10}$	$4.24\times 10^{14}$	$1.11\times 10^{15}$	$1.53\times 10^{9}$	$2.21\times 10^{9}$
$f_{4}$	20	$7.39\times 10^{3}$	$5.27\times 10^{3}$	$4.42\times 10^{4}$	$1.24\times 10^{4}$	$7.34\times 10^{4}$	$2.67\times 10^{4}$	$4.59\times 10^{4}$	$9.36\times 10^{3}$	$3.29\times 10^{4}$	$1.21\times 10^{4}$
	30	$1.85\times 10^{4}$	$7.33\times 10^{3}$	$7.05\times 10^{4}$	$1.16\times 10^{4}$	$1.02\times 10^{5}$	$3.57\times 10^{4}$	$6.93\times 10^{4}$	$9.76\times 10^{3}$	$5.89\times 10^{4}$	$1.17\times 10^{4}$
	40	$2.54\times 10^{4}$	$8.33\times 10^{3}$	$8.38\times 10^{4}$	$1.50\times 10^{4}$	$1.09\times 10^{5}$	$3.20\times 10^{4}$	$8.61\times 10^{4}$	$1.18\times 10^{4}$	$6.98\times 10^{4}$	$1.72\times 10^{4}$
$f_{5}$	20	$-8.95\times 10^{2}$	$3.60\times 10^{2}$	$-1.00\times 10^{3}$	$1.43\times 10^{-1}$	$-6.54\times 10^{2}$	$1.76\times 10^{2}$	$5.49\times 10^{2}$	$3.12\times 10^{2}$	$-1.00\times 10^{3}$	$1.71\times 10^{1}$
	30	$-8.20\times 10^{2}$	$3.86\times 10^{2}$	$-9.76\times 10^{2}$	$3.42\times 10^{1}$	$-2.32\times 10^{1}$	$2.11\times 10^{2}$	$2.48\times 10^{3}$	$5.86\times 10^{2}$	$-9.98\times 10^{2}$	$2.75\times 10^{1}$
	40	$-5.96\times 10^{2}$	$6.68\times 10^{2}$	$-9.31\times 10^{2}$	$3.06\times 10^{1}$	$4.51\times 10^{2}$	$2.51\times 10^{2}$	$2.92\times 10^{3}$	$4.90\times 10^{2}$	$-9.75\times 10^{2}$	$3.70\times 10^{1}$

**Table 2 entropy-24-01109-t002:** The experimental results under composition test functions.

Algorithm		PSO		AFMO		WOA		BH		GAFMO	
	Dim	Ave	Std	Ave	Std	Ave	Std	Ave	Std	Ave	Std
$f_{21}$	20	$1.04\times 10^{3}$	$8.38\times 10^{1}$	$1.08\times 10^{3}$	$4.07\times 10^{1}$	$1.41\times 10^{3}$	$2.38\times 10^{2}$	$1.80\times 10^{3}$	$3.63\times 10^{1}$	$1.03\times 10^{3}$	$3.16\times 10^{1}$
	30	$1.06\times 10^{3}$	$8.70\times 10^{1}$	$1.07\times 10^{3}$	$7.46\times 10^{1}$	$1.74\times 10^{3}$	$4.16\times 10^{2}$	$2.74\times 10^{3}$	$5.24\times 10^{1}$	$1.05\times 10^{3}$	$5.32\times 10^{1}$
	40	$1.41\times 10^{3}$	$6.26\times 10^{1}$	$1.39\times 10^{3}$	$9.19\times 10^{1}$	$2.18\times 10^{3}$	$3.71\times 10^{2}$	$3.66\times 10^{3}$	$1.10\times 10^{2}$	$1.41\times 10^{3}$	$4.81\times 10^{1}$
$f_{22}$	20	$3.92\times 10^{3}$	$6.99\times 10^{2}$	$5.93\times 10^{3}$	$1.82\times 10^{2}$	$4.88\times 10^{3}$	$5.64\times 10^{2}$	$5.65\times 10^{3}$	$4.20\times 10^{2}$	$2.48\times 10^{3}$	$3.56\times 10^{2}$
	30	$5.90\times 10^{3}$	$1.08\times 10^{3}$	$9.00\times 10^{3}$	$2.65\times 10^{2}$	$8.22\times 10^{3}$	$7.99\times 10^{2}$	$8.61\times 10^{3}$	$6.79\times 10^{2}$	$3.71\times 10^{3}$	$5.75\times 10^{2}$
	40	$9.24\times 10^{3}$	$9.69\times 10^{2}$	$1.31\times 10^{4}$	$2.92\times 10^{2}$	$1.17\times 10^{4}$	$9.10\times 10^{2}$	$1.27\times 10^{4}$	$6.73\times 10^{2}$	$6.09\times 10^{3}$	$6.62\times 10^{2}$
$f_{23}$	20	$4.28\times 10^{3}$	$6.47\times 10^{2}$	$5.98\times 10^{3}$	$2.47\times 10^{2}$	$5.38\times 10^{3}$	$5.48\times 10^{2}$	$5.74\times 10^{3}$	$4.88\times 10^{2}$	$3.86\times 10^{3}$	$3.63\times 10^{2}$
	30	$6.43\times 10^{3}$	$8.95\times 10^{2}$	$9.53\times 10^{3}$	$3.30\times 10^{2}$	$8.33\times 10^{3}$	$6.61\times 10^{2}$	$8.72\times 10^{3}$	$8.73\times 10^{2}$	$6.45\times 10^{3}$	$4.19\times 10^{2}$
	40	$9.31\times 10^{3}$	$1.14\times 10^{3}$	$1.32\times 10^{4}$	$3.37\times 10^{2}$	$1.21\times 10^{4}$	$9.08\times 10^{2}$	$1.29\times 10^{4}$	$6.63\times 10^{2}$	$7.86\times 10^{3}$	$6.08\times 10^{2}$
$f_{24}$	20	$1.25\times 10^{3}$	$6.56\times 10^{0}$	$1.26\times 10^{3}$	$7.20\times 10^{0}$	$1.28\times 10^{3}$	$8.62\times 10^{0}$	$1.29\times 10^{3}$	$1.31\times 10^{1}$	$1.24\times 10^{3}$	$5.73\times 10^{0}$
	30	$1.29\times 10^{3}$	$1.02\times 10^{1}$	$1.30\times 10^{3}$	$8.98\times 10^{0}$	$1.32\times 10^{3}$	$1.14\times 10^{1}$	$1.36\times 10^{3}$	$2.09\times 10^{1}$	$1.27\times 10^{3}$	$7.30\times 10^{0}$
	40	$1.33\times 10^{3}$	$1.48\times 10^{1}$	$1.34\times 10^{3}$	$1.34\times 10^{1}$	$1.37\times 10^{3}$	$1.45\times 10^{1}$	$1.46\times 10^{3}$	$2.39\times 10^{1}$	$1.30\times 10^{3}$	$8.88\times 10^{0}$
$f_{25}$	20	$1.37\times 10^{3}$	$9.91\times 10^{0}$	$1.37\times 10^{3}$	$6.68\times 10^{0}$	$1.38\times 10^{3}$	$7.58\times 10^{0}$	$1.41\times 10^{3}$	$8.95\times 10^{0}$	$1.36\times 10^{3}$	$7.87\times 10^{0}$
	30	$1.41\times 10^{3}$	$1.16\times 10^{1}$	$1.42\times 10^{3}$	$6.86\times 10^{0}$	$1.43\times 10^{3}$	$1.21\times 10^{1}$	$1.49\times 10^{3}$	$1.27\times 10^{1}$	$1.39\times 10^{3}$	$6.82\times 10^{0}$
	40	$1.49\times 10^{3}$	$2.43\times 10^{1}$	$1.50\times 10^{3}$	$1.54\times 10^{1}$	$1.50\times 10^{3}$	$1.35\times 10^{1}$	$1.62\times 10^{3}$	$1.90\times 10^{1}$	$1.44\times 10^{3}$	$1.17\times 10^{1}$
$f_{26}$	20	$1.47\times 10^{3}$	$7.21\times 10^{1}$	$1.42\times 10^{3}$	$5.06\times 10^{1}$	$1.51\times 10^{3}$	$7.57\times 10^{1}$	$1.41\times 10^{3}$	$1.75\times 10^{0}$	$1.50\times 10^{3}$	$4.69\times 10^{1}$
	30	$1.54\times 10^{3}$	$6.42\times 10^{1}$	$1.48\times 10^{3}$	$8.50\times 10^{1}$	$1.59\times 10^{3}$	$5.62\times 10^{1}$	$1.44\times 10^{3}$	$6.26\times 10^{1}$	$1.53\times 10^{3}$	$8.45\times 10^{1}$
	40	$1.58\times 10^{3}$	$6.32\times 10^{1}$	$1.54\times 10^{3}$	$8.96\times 10^{1}$	$1.60\times 10^{3}$	$9.48\times 10^{1}$	$1.59\times 10^{3}$	$9.26\times 10^{1}$	$1.54\times 10^{3}$	$8.74\times 10^{1}$
$f_{27}$	20	$2.06\times 10^{3}$	$6.37\times 10^{1}$	$2.13\times 10^{3}$	$5.10\times 10^{1}$	$2.27\times 10^{3}$	$6.91\times 10^{1}$	$2.31\times 10^{3}$	$7.61\times 10^{1}$	$1.99\times 10^{3}$	$6.66\times 10^{1}$
	30	$2.38\times 10^{3}$	$1.04\times 10^{2}$	$2.50\times 10^{3}$	$6.59\times 10^{1}$	$2.70\times 10^{3}$	$8.63\times 10^{1}$	$2.74\times 10^{3}$	$1.03\times 10^{2}$	$2.26\times 10^{3}$	$4.78\times 10^{1}$
	40	$2.75\times 10^{3}$	$1.23\times 10^{2}$	$2.89\times 10^{3}$	$7.69\times 10^{1}$	$3.16\times 10^{3}$	$1.19\times 10^{2}$	$3.45\times 10^{3}$	$1.23\times 10^{2}$	$2.48\times 10^{3}$	$6.32\times 10^{1}$
$f_{28}$	20	$2.94\times 10^{3}$	$8.24\times 10^{2}$	$3.62\times 10^{3}$	$5.02\times 10^{2}$	$5.70\times 10^{3}$	$7.39\times 10^{2}$	$5.17\times 10^{3}$	$4.43\times 10^{2}$	$2.36\times 10^{3}$	$3.67\times 10^{2}$
	30	$2.52\times 10^{3}$	$7.67\times 10^{2}$	$2.59\times 10^{3}$	$1.04\times 10^{3}$	$6.36\times 10^{3}$	$6.83\times 10^{2}$	$6.08\times 10^{3}$	$4.78\times 10^{2}$	$2.04\times 10^{3}$	$2.87\times 10^{2}$
	40	$3.59\times 10^{3}$	$9.24\times 10^{2}$	$3.19\times 10^{3}$	$1.02\times 10^{3}$	$7.80\times 10^{3}$	$1.14\times 10^{3}$	$8.41\times 10^{3}$	$6.37\times 10^{2}$	$2.69\times 10^{3}$	$2.57\times 10^{2}$

**Table 3 entropy-24-01109-t003:** The experimental results under multi-modal test functions.

Algorithm		PSO		AFMO		WOA		BH		GAFMO	
	Dim	Ave	Std	Ave	Std	Ave	Std	Ave	Std	Ave	Std
$f_{6}$	20	$-8.35\times 10^{2}$	$3.33\times 10^{1}$	$-8.62\times 10^{2}$	$3.07\times 10^{1}$	$-7.50\times 10^{2}$	$6.53\times 10^{1}$	$3.43\times 10^{2}$	$2.28\times 10^{2}$	$-8.94\times 10^{2}$	$1.58\times 10^{1}$
	30	$-7.96\times 10^{2}$	$5.52\times 10^{1}$	$-8.43\times 10^{2}$	$3.45\times 10^{1}$	$-5.67\times 10^{2}$	$1.33\times 10^{2}$	$1.52\times 10^{3}$	$3.37\times 10^{2}$	$-8.52\times 10^{2}$	$2.60\times 10^{1}$
	40	$-7.48\times 10^{2}$	$4.23\times 10^{1}$	$-7.97\times 10^{2}$	$3.23\times 10^{1}$	$-4.47\times 10^{2}$	$1.06\times 10^{2}$	$1.48\times 10^{3}$	$2.25\times 10^{2}$	$-8.08\times 10^{2}$	$3.93\times 10^{1}$
$f_{7}$	20	$-7.43\times 10^{2}$	$3.34\times 10^{1}$	$-6.65\times 10^{2}$	$1.29\times 10^{2}$	$3.10\times 10^{3}$	$8.46\times 10^{3}$	$3.34\times 10^{4}$	$2.66\times 10^{4}$	$-7.78\times 10^{2}$	$6.67\times^{0}$
	30	$-6.83\times 10^{2}$	$3.82\times 10^{1}$	$-6.35\times 10^{2}$	$6.41\times 10^{1}$	$9.56\times 10^{2}$	$4.98\times 10^{3}$	$5.85\times 10^{4}$	$1.55\times 10^{5}$	$-7.15\times 10^{2}$	$2.02\times 10^{1}$
	40	$-6.70\times 10^{2}$	$5.17\times 10^{1}$	$-6.22\times 10^{2}$	$5.81\times 10^{1}$	$-4.16\times 10^{2}$	$2.59\times 10^{2}$	$2.19\times 10^{3}$	$3.31\times 10^{3}$	$-7.22\times 10^{2}$	$1.92\times 10^{1}$
$f_{8}$	20	$-6.79\times 10^{2}$	$7.18\times 10^{-2}$	$-6.79\times 10^{2}$	$6.40\times 10^{-2}$	$-6.79\times 10^{2}$	$7.44\times 10^{-2}$	$-6.79\times 10^{2}$	$7.36\times 10^{-2}$	$-6.79\times 10^{2}$	$8.27\times 10^{-2}$
	30	$-6.79\times 10^{2}$	$7.02\times 10^{-2}$	$-6.79\times 10^{2}$	$5.15\times 10^{-2}$	$-6.79\times 10^{2}$	$6.80\times 10^{-2}$	$-6.79\times 10^{2}$	$6.44\times 10^{-2}$	$-6.79\times 10^{2}$	$5.21\times 10^{-2}$
	40	$-6.79\times 10^{2}$	$7.62\times 10^{-2}$	$-6.79\times 10^{2}$	$3.68\times 10^{-2}$	$-6.79\times 10^{2}$	$8.00\times 10^{-2}$	$-6.79\times 10^{2}$	$5.71\times 10^{-2}$	$-6.79\times 10^{2}$	$4.11\times 10^{-2}$
$f_{9}$	20	$-5.85\times 10^{2}$	$3.18\times 10^{0}$	$-5.79\times 10^{2}$	$1.51\times 10^{0}$	$-5.77\times 10^{2}$	$2.09\times 10^{0}$	$-5.77\times 10^{2}$	$2.90\times 10^{0}$	$-5.89\times 10^{2}$	$1.52\times 10^{0}$
	30	$-5.71\times 10^{2}$	$3.33\times 10^{0}$	$-5.60\times 10^{2}$	$1.59\times 10^{0}$	$-5.62\times 10^{2}$	$2.90\times 10^{0}$	$-5.61\times 10^{2}$	$2.57\times 10^{0}$	$-5.79\times 10^{2}$	$1.93\times 10^{0}$
	40	$-5.61\times 10^{2}$	$4.14\times 10^{0}$	$-5.47\times 10^{2}$	$2.02\times 10^{0}$	$-5.46\times 10^{2}$	$3.59\times 10^{0}$	$-5.46\times 10^{2}$	$3.15\times 10^{0}$	$-5.71\times 10^{2}$	$2.03\times 10^{0}$
$f_{10}$	20	$-4.58\times 10^{2}$	$3.95\times 10^{1}$	$-4.67\times 10^{2}$	$3.10\times 10^{1}$	$-2.77\times 10^{2}$	$1.08\times 10^{2}$	$1.76\times 10^{2}$	$8.47\times 10^{1}$	$-4.98\times 10^{2}$	$4.74\times 10^{0}$
	30	$-3.71\times 10^{2}$	$1.34\times 10^{2}$	$-3.81\times 10^{2}$	$9.33\times 10^{1}$	$1.23\times 10^{2}$	$2.20\times 10^{2}$	$2.00\times 10^{3}$	$2.18\times 10^{2}$	$-4.96\times 10^{2}$	$1.42\times 10^{1}$
	40	$-3.93\times 10^{2}$	$1.28\times 10^{2}$	$-1.24\times 10^{2}$	$3.03\times 10^{2}$	$6.42\times 10^{2}$	$3.11\times 10^{2}$	$2.02\times 10^{3}$	$1.95\times 10^{2}$	$-4.93\times 10^{2}$	$3.91\times 10^{1}$
$f_{11}$	20	$-3.26\times 10^{2}$	$2.43\times 10^{1}$	$-2.55\times 10^{2}$	$1.30\times 10^{1}$	$-9.46\times 10^{1}$	$7.10\times 10^{1}$	$-1.44\times 10^{2}$	$4.14\times 10^{1}$	$-3.63\times 10^{2}$	$1.30\times 10^{1}$
	30	$-2.20\times 10^{2}$	$6.33\times 10^{1}$	$-1.43\times 10^{2}$	$2.85\times 10^{1}$	$1.63\times 10^{2}$	$1.07\times 10^{2}$	$1.23\times 10^{2}$	$6.70\times 10^{1}$	$-3.24\times 10^{2}$	$.25\times 10^{1}$
	40	$-6.91\times 10^{1}$	$7.61\times 10^{1}$	$-7.11\times 10^{0}$	$4.47\times 10^{1}$	$3.82\times 10^{2}$	$1.07\times 10^{2}$	$3.35\times 10^{2}$	$1.02\times 10^{2}$	$-2.55\times 10^{2}$	$2.94\times 10^{1}$
$f_{12}$	20	$-2.10\times 10^{2}$	$3.23\times 10^{1}$	$-1.37\times 10^{2}$	$1.72\times 10^{1}$	$-9.94\times 10^{0}$	$7.73\times 10^{1}$	$-5.64\times 10^{0}$	$5.32\times 10^{1}$	$-2.28\times 10^{2}$	$1.50\times 10^{1}$
	30	$-1.25\times 10^{2}$	$5.74\times 10^{1}$	$3.25\times 10^{1}$	$3.13\times 10^{1}$	$2.94\times 10^{2}$	$1.15\times 10^{2}$	$2.17\times 10^{2}$	$6.76\times 10^{1}$	$-1.40\times 10^{2}$	$1.86\times 10^{1}$
	40	$9.51\times 10^{0}$	$7.05\times 10^{1}$	$2.10\times 10^{2}$	$2.96\times 10^{1}$	$5.74\times 10^{2}$	$1.07\times 10^{2}$	4.32$\times 10^{2}$	7.85$\times 10^{1}$	$-5.07\times 10^{1}$	$2.81\times 10^{1}$
$f_{13}$	20	$-7.14\times 10^{1}$	$2.57\times 10^{1}$	$-3.15\times 10^{1}$	$1.60\times 10^{1}$	$7.73\times 10^{1}$	$7.23\times 10^{1}$	$1.17\times 10^{2}$	$5.24\times 10^{1}$	$-9.70\times 10^{1}$	$1.32\times 10^{1}$
	30	$4.18\times 10^{1}$	$4.87\times 10^{1}$	$1.34\times 10^{2}$	$2.69\times 10^{1}$	$3.71\times 10^{2}$	$1.13\times 10^{2}$	$3.46\times 10^{2}$	$5.91\times 10^{1}$	$1.10\times 10^{1}$	$1.75\times 10^{1}$
	40	$1.76\times 10^{2}$	$7.05\times 10^{1}$	$3.16\times 10^{2}$	$3.66\times 10^{1}$	$6.58\times 10^{2}$	$1.05\times 10^{2}$	$6.04\times 10^{2}$	$7.87\times 10^{1}$	$1.08\times 10^{2}$	$3.20\times 10^{1}$
$f_{14}$	20	$2.00\times 10^{3}$	$4.38\times 10^{2}$	$4.36\times 10^{3}$	$2.60\times 10^{2}$	$3.27\times 10^{3}$	$5.94\times 10^{2}$	$4.02\times 10^{3}$	$5.35\times 10^{2}$	$1.00\times 10^{3}$	$3.56\times 10^{2}$
	30	$3.67\times 10^{3}$	$6.30\times 10^{2}$	$7.49\times 10^{3}$	$2.74\times 10^{2}$	$5.81\times 10^{3}$	$6.84\times 10^{2}$	$6.95\times 10^{3}$	$7.51\times 10^{2}$	$2.40\times 10^{3}$	$5.59\times 10^{2}$
	40	$5.62\times 10^{3}$	$7.67\times 10^{2}$	$1.07\times 10^{4}$	$3.21\times 10^{2}$	$8.30\times 10^{3}$	$8.81\times 10^{2}$	$1.01\times 10^{4}$	$8.51\times 10^{2}$	$3.92\times 10^{3}$	$6.02\times 10^{2}$
$f_{15}$	20	$2.25\times 10^{3}$	$5.08\times 10^{2}$	$4.39\times 10^{3}$	$2.64\times 10^{2}$	$3.53\times 10^{3}$	$5.67\times 10^{2}$	$3.81\times 10^{3}$	$6.77\times 10^{2}$	$2.33\times 10^{3}$	$3.13\times 10^{2}$
	30	$4.28\times 10^{3}$	$7.14\times 10^{2}$	$7.80\times 10^{3}$	$2.69\times 10^{2}$	$6.57\times 10^{3}$	$8.39\times 10^{2}$	$7.11\times 10^{3}$	$8.37\times 10^{2}$	$4.82\times 10^{3}$	$3.46\times 10^{2}$
	40	$6.43\times 10^{3}$	$8.70\times 10^{2}$	$1.13\times 10^{4}$	$3.24\times 10^{2}$	$9.27\times 10^{3}$	$9.20\times 10^{2}$	$1.06\times 10^{4}$	$7.27\times 10^{2}$	$5.97\times 10^{3}$	$4.32\times 10^{2}$
$f_{16}$	20	$2.01\times 10^{2}$	$4.78\times 10^{-1}$	$2.02\times 10^{2}$	$3.40\times 10^{-1}$	$2.02\times 10^{2}$	$5.20\times 10^{-1}$	$2.02\times 10^{2}$	$4.66\times 10^{-1}$	$2.02\times 10^{2}$	$2.89\times 10^{-1}$
	30	$2.02\times 10^{2}$	$5.85\times 10^{-1}$	$2.03\times 10^{2}$	$3.23\times 10^{-1}$	$2.02\times 10^{2}$	$6.65\times 10^{-1}$	$2.02\times 10^{2}$	$5.14\times 10^{-1}$	$2.03\times 10^{2}$	$3.75\times 10^{-1}$
	40	$2.03\times 10^{2}$	$6.60\times 10^{-1}$	$2.04\times 10^{2}$	$3.06\times 10^{-1}$	$2.03\times 10^{2}$	$5.88\times 10^{-1}$	$2.03\times 10^{2}$	$5.73\times 10^{-1}$	$2.03\times 10^{2}$	$4.39\times 10^{-1}$
$f_{17}$	20	$3.77\times 10^{2}$	$1.33\times 10^{1}$	$5.18\times 10^{2}$	$1.80\times 10^{1}$	$6.58\times 10^{2}$	$7.42\times 10^{1}$	$5.74\times 10^{2}$	$4.23\times 10^{1}$	$4.34\times 10^{2}$	$1.60\times 10^{1}$
	30	$4.71\times 10^{2}$	$2.47\times 10^{1}$	$7.03\times 10^{2}$	$2.79\times 10^{1}$	$9.95\times 10^{2}$	$1.09\times 10^{2}$	$8.44\times 10^{2}$	$8.01\times 10^{1}$	$5.49\times 10^{2}$	$2.74\times 10^{1}$
	40	$5.88\times 10^{2}$	$3.95\times 10^{1}$	$9.15\times 10^{2}$	$3.28\times 10^{1}$	$1.29\times 10^{3}$	$1.02\times 10^{2}$	$1.11\times 10^{3}$	$1.23\times 10^{2}$	$6.95\times 10^{2}$	$3.21\times 10^{1}$
$f_{18}$	20	$4.93\times 10^{2}$	$1.95\times 10^{1}$	$6.11\times 10^{2}$	$1.63\times 10^{1}$	$7.54\times 10^{2}$	$7.07\times 10^{1}$	$6.72\times 10^{2}$	$6.48\times 10^{1}$	$5.50\times 10^{2}$	$1.40\times 10^{1}$
	30	$5.93\times 10^{2}$	$3.04\times 10^{1}$	$8.05\times 10^{2}$	$2.40\times 10^{1}$	$1.10\times 10^{3}$	$1.12\times 10^{2}$	$9.54\times 10^{2}$	$7.99\times 10^{1}$	$6.83\times 10^{2}$	$1.87\times 10^{1}$
	40	$7.00\times 10^{2}$	$4.02\times 10^{1}$	$1.03\times 10^{3}$	$3.84\times 10^{1}$	$1.43\times 10^{3}$	$1.17\times 10^{2}$	$1.24\times 10^{3}$	$1.06\times 10^{2}$	$8.27\times 10^{2}$	$2.72\times 10^{1}$
$f_{19}$	20	$5.58\times 10^{2}$	$2.78\times 10^{2}$	$5.06\times 10^{2}$	$1.96\times 10^{0}$	$5.50\times 10^{2}$	$2.59\times 10^{1}$	$1.80\times 10^{3}$	$4.13\times 10^{2}$	$5.05\times 10^{2}$	$1.38\times 10^{0}$
	30	$5.15\times 10^{2}$	$1.16\times 10^{1}$	$5.13\times 10^{2}$	$3.53\times 10^{0}$	$7.59\times 10^{2}$	$2.00\times 10^{2}$	$1.54\times 10^{4}$	$3.45\times 10^{3}$	$5.11\times 10^{2}$	$2.47\times 10^{0}$
	40	$7.29\times 10^{2}$	$6.18\times 10^{2}$	$5.23\times 10^{2}$	$4.66\times 10^{0}$	$1.57\times 10^{3}$	$1.24\times 10^{3}$	$4.38\times 10^{4}$	$1.03\times 10^{4}$	$5.19\times 10^{2}$	$3.31\times 10^{0}$
$f_{20}$	20	$6.09\times 10^{2}$	$6.54\times 10^{-1}$	$6.10\times 10^{2}$	$5.77\times 10^{-10}$	$6.10\times 10^{2}$	$1.70\times 10^{-1}$	$6.10\times 10^{2}$	$1.47\times 10^{-1}$	$6.10\times 10^{2}$	$9.66\times 10^{-1}$
	30	$6.15\times 10^{2}$	$8.75\times 10^{-1}$	$6.15\times 10^{2}$	$3.30\times 10^{-7}$	$6.15\times 10^{2}$	$1.92\times 10^{-1}$	$6.15\times 10^{2}$	$1.70\times 10^{-1}$	$6.15\times 10^{2}$	$9.60\times 10^{-1}$
	40	$6.18\times 10^{2}$	$6.34\times 10^{-1}$	$6.19\times 10^{2}$	$1.48\times 10^{-1}$	$6.19\times 10^{2}$	$4.07\times 10^{-1}$	$6.18\times 10^{2}$	$4.08\times 10^{-1}$	$6.18\times 10^{2}$	$2.28\times 10^{-1}$

**Table 4 entropy-24-01109-t004:** The simulation results under different anchor node number.

Anchor Node Number	sMCL	BH	PSO	WOA	AFMO	GAFMO
A = 5	35.9050	25.1668	24.8511	24.9971	24.8520	20.0493
A = 10	21.5088	11.5516	11.3086	11.4438	11.3504	9.0480
A = 15	18.5468	9.5519	9.3780	9.4786	9.3793	7.6191
A = 20	13.1348	5.1515	5.0186	5.1158	5.0014	3.8387
A = 25	11.4025	3.8816	3.7762	3.8889	3.7793	3.0388
A = 30	13.4114	5.8577	5.7544	5.8850	5.7661	4.5912

**Table 5 entropy-24-01109-t005:** The simulation results under different sensor node number.

Sensor Node Number	sMCL	BH	PSO	WOA	AFMO	GAMFO
N = 50	36.1180	26.2782	25.8972	25.9487	25.8977	20.3007
N = 100	23.2155	13.2985	13.0395	13.1358	13.0415	10.5978
N = 150	23.2896	13.3750	13.1198	13.2913	13.1536	10.2857
N = 200	21.5088	11.5516	11.3086	11.4438	11.3504	9.0480
N = 250	25.4070	15.5492	15.4005	15.4271	15.3327	12.4996
N = 300	28.1844	18.3840	18.0993	18.2259	18.0933	15.6258

**Table 6 entropy-24-01109-t006:** The simulation results under different communication radii.

Communication Radius	sMCL	BH	PSO	WOA	AFMO	GAMO
R = 15	25.3064	14.9851	14.6533	14.7383	14.6083	12.2787
R = 20	20.6724	10.6305	10.3826	10.5019	10.3850	8.5380
R = 25	26.8183	17.0337	16.7616	16.8847	16.7640	13.8482
R = 30	20.5605	11.0756	10.8769	10.9854	10.8813	9.0985
R = 35	17.8197	8.9234	8.7262	8.8501	8.7289	7.0821
R = 40	13.1540	5.0344	4.9078	5.0095	4.9114	4.1041

## Data Availability

Not applicable.

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
