# Peer review of "Gaussian-Based Adaptive Fish Migration Optimization Applied to Optimization Localization Error of Mobile Sensor Networks"

_entropy, 2022, doi:10.3390/e24081109_

Round 1

Reviewer 1 Report

The paper presents an algorithm for localization of sensor nodes in wireless networks with improved accuracy. It uses the Adaptive fish migration optimization algorithm in conjunction with the Gaussian noise in order to generate new individual in migration process of AFMO. The results are claimed to bring better performances and the ability to avoid the optimal local value.

Some suggestions about the way the paper is organized:

- Figure 2 is located before its call in text.

- Figure 3: "The Mobile Sensor None" - is it correct?

- Table 1, 2, and 3 are located before their call in the text

- "The combination function combines uni-modal and multi-modal test functions" - combination and combine together

- Tables 1-6 shows some numbers, a lot of numbers for tables 1-3, but it is not clear or difficult to understand what they represents (absolute error, relative error or position). They are not well commented and explained in the text.

It is not clear which is the connection between the results offered by the test functions used in 4.1. and the resulting error for the localisation of mobile nodes. As stated, the heuristic methods are usually used to solve a problem in a faster way while sacrificing accuracy. It might be useful for the reader to have some comments about this aspect.

Somewhere in the paper the authors speak about the speed of the nodes. Is this parameter analyzed by the authors? Does it influences the localization error?

Reviewer 2 Report

Add in the abstract the explicit aim of the proposed study and its improvements. I think the authors should report significant results in the abstract to promote the performance of their proposed method.

The section Introduction should clarify better and provide concise information with regard to the problem definition and scope of the paper. The contribution summarization should be remarked better. Moreover, the connection between the problem and the solution proposed is also not pointed out clearly. Emphasize the novelty introduced.

About the related works, further papers should be added to the literature review. Each paper should clearly specify what is the proposed methodology, novelty, and results with experimentation. At the end of related works, highlight in some lines what overall technical gaps are observed in existing works, that led to the design of the proposed approach. To better delineate the context and the different possible solutions, you can consider the following papers as references: "A Multi Agent Approach for the Construction of a Peer-to-Peer Information System in Grids", Self-Organ. Auton. Inform.(I) and "Heuristic recommendation technique in Internet of Things featuring swarm intelligence approach", ESWA.

The future scope of the methodology should be extended/highlighted. Improve the conclusion, and clarify the conclusion of this article with its significance for follow-up research.

Round 2

Reviewer 2 Report

To better delineate the context and the different possible solutions, suggestions provided in the previous review should be considered. The conclusion should be further improved. The authors should focus on their unique work and contributions at first, and they should support their conclusion with numerical results. Then, the limitations of this paper should be discussed. Accordingly, the future work of this paper can be drawn.

Author Response

Thanks very much for your advice. The conclusion section has been modified and added some content as you suggested.